

# Assessment of liquid media requirements for storing and evaluating respiratory cilia motility

Richard Francis

College of Medicine and Dentistry, James Cook University, Townsville, QLD, Australia

Corresponding author
Richard Francis,
richard.francis@jcu.edu.au

## ABSTRACT

Mucociliary clearance is critical for maintaining normal lung function. Respiratory cilia which drive mucociliary clearance are commonly studied by measuring cilia beat frequency (CBF). There is currently significant variation within the literature regarding what is a normal value for CBF, this may be due in part to the large variety of liquid media used to suspend, maintain, and image ciliated cells. This study aimed to conduct a thorough examination to assess how media choice influences respiratory cilia motility. To accomplish this, Adult C57/BL6 mouse trachea samples were incubated in eight commonly used liquid media including: Saline, Dulbecco's Phosphate-Buffered Saline (DPBS), Hanks' Balanced Salt Solution (HBSS), Medium 199 (M199), Dulbecco's Modified Eagle's Medium (DMEM), Roswell Park Memorial Institute Medium (RPMI), Minimum Essential Medium (MEM), and Leibovitz's L-15 Medium (L-15); with or without 10% FBS supplementation. The effects of storage time (0–12 hours) and storage temperature (4 °C or room temperature) were also assessed. All media except saline were found to be equally effective in maintaining cilia function in airway samples that were freshly harvested and immediately imaged. Saline, however, significantly reduced the number of cells with motile cilia. A more complex pattern emerged when samples were stored before imaging. In saline, cilia function was significantly impaired after just one hour of storage. Samples stored in all other media showed strong maintenance of motile cilia function, with only minor changes. Notably, cilia function was better preserved with storage at 4 °C, while room temperature storage generally led to significant increases in CBF, especially in media containing FBS. Lastly, FBS supplementation was essential for maintaining cilia motility in L-15 media, as L-15 without FBS resulted in significant decreases in cilia motility following storage at either 4 °C or room temperature. In conclusion, saline should only be used if cilia are to be imaged immediately, as cilia stored in saline quickly lose motile function. All other commonly used media appear equally capable of maintaining motile cilia function for up to 12 hours when stored at 4 °C. Surprisingly, DPBS was just as effective as more expensive media in preserving ciliated samples. Storing ciliated tissue at room temperature generally leads to increased CBF, particularly in media containing FBS. Finally, L-15 media alone specifically requires the addition of 10% FBS to maintain cilia motility. These findings provide a valuable foundation for standardizing the handling, collection, and transport of ciliated samples for motile cilia assessment.

## INTRODUCTION

The epithelial surface of the conducting airways are covered by motile cilia that beat in a coordinated manner to clear the lungs of mucus and trapped inhaled foreign particles in a process called mucociliary clearance (*Bustamante-Marin & Ostrowski, 2017*). Respiratory cilia function is commonly studied by measuring factors influencing cilia beat frequency (CBF) (*Bauer et al., 2024*; *Bishop et al., 2024*; *Christopher et al., 2014*; *Welchering et al., 2015*). However, one problem with past cilia studies is the reported large variation in control CBFs, ranging between 7–20 Hz (*Fawcett et al., 2023*; *Price et al., 2015*; *Zahid et al., 2020*; *Scopulovic et al., 2022*). While this large heterogeneity in control CBF may be related to experimental model used (animal *vs* human; fresh *vs* cultured tissue) one confounding variable preventing easy reconciliation of published CBF data, even between studies using similar model systems, is the different types of liquid media used to suspend, maintain, and image ciliated cells. A wide range of media have been previously used to study cilia function, including phosphate-buffered saline (PBS) (*Mateos-Quiros et al., 2021*), Hanks' balanced salt solution (HBSS) (*Chen, Lemieux & Wong, 2016*; *Jing et al., 2017*), Leibovitz's L-15 medium (L-15) (*Scopulovic et al., 2022*; *Zahid et al., 2020*), Medium 199 (M199) (*Bauer et al., 2024*; *Bricmont et al., 2023*; *Price et al., 2015*), Roswell Park Memorial Institute medium (RPMI) (*Behr et al., 2023*; *Bishop et al., 2024*), and Dulbecco's modified eagle's medium (DMEM) (*Lever et al., 2024*; *Reula et al., 2021*). Supplementation of media also varies between studies, the most common difference being the addition or omission of fetal bovine serum (FBS), as highlighted in two recent studies using RPMI (*Behr et al., 2023*; *Bishop et al., 2024*).

The importance of assessing cilia motility is easy to appreciate by considering diseases where this motility is impaired. Primary ciliary dyskinesia (PCD) is a genetic disorder where gene mutations cause immotile or highly dyskinetic cilia and abnormal mucociliary clearance (*Raidt et al., 2023*). PCD patients often exhibit lung disease at birth with neonatal respiratory distress followed by a lifetime of sinopulmonary disease that can eventually require lung transplantation (*Despotes et al., 2024*). Diagnosis of PCD should logically be easy, requiring the simple assessment of respiratory cilia motility *via* high-speed microscopy analysis (HSMA) using easily collected nasal epithelial brushings (fresh or cultured samples *Muller et al., 2021*). However, while HSMA is a recognized method of PCD diagnosis (*Despotes et al., 2024*; *Shapiro et al., 2018*), it is not considered a gold standard for diagnosis of PCD because it requires highly specialized instruments/personal and has not yet been standardized and validated (*Despotes et al., 2024*). One item needing standardization is the type of cell media used for the harvest/imaging of ciliated samples, in addition to the effect of transport (*i.e.,* storage temperature/time) of samples to locations for HSMA.

Thus, the aim of this study was to firstly determine if culture media choice significantly influences respiratory cilia motile function by conducting a thorough systematic examination of cilia motility within cell media commonly used to study cilia function, including the effect of FBS supplementation. Secondly, to test the effect of storing ciliated samples in each media at different temperatures/times to aid in standardization of handling

protocols for collection and transport of ciliated tissue samples to specialized laboratories for PCD diagnostic testing.

## MATERIALS & METHODS

### Animals

All animal procedures were conducted in accordance with the James Cook University Animal Ethics Committee (Ethics# A2783). Adult C57/BL6 mice aged between 12–24 months old destined for euthanasia during routine colony maintenance were donated to this study by the Australian Institute of Tropical Health & Medicine small animal colony. The animals were housed in an air-conditioned room equipped with racked mouse cages connected to a Smart Flow ventilation system (Tecniplast, Buguggiate, Italy). They had free access to standard rodent chow and water. To enrich their environment, the cages were provided with various bedding materials, including sawdust and shredded paper, as well as cardboard tubes for hiding and sleeping. New mice were delivered weekly and euthanized within the same week using carbon dioxide asphyxiation (five L/min $CO_2$ gas using a 7L euthanasia chamber). Euthanasia was verified by the absence of corneal and toe-pinch reflexes. All mice delivered for this study were euthanized and utilized for data collection.

### Liquid media

Eight common liquid media used to store/study ciliated samples were tested in this study (Table 1). Except for saline, all media were supplemented with penicillin G sodium (100 units/ml) and streptomycin sulfate (100 µg/ml) (15140122; Thermo Fisher Scientific). All media (except for saline) were tested with or without 10% FBS (16000036; Thermo Fisher Scientific) as described.

### Airway tissue harvest and sample preparation

Ciliated trachea epithelial samples were harvested as previously described (*Francis & Lo, 2013*). In brief, mice were euthanised by $CO_2$ asphyxiation and trachea were immediately harvested and placed into one of the eight liquid media being tested (Table 1; on ice, with or without FBS). Fine dissection was then used to prepare four tissue samples from each trachea, each being 3–4 tracheal rings in length. Division of each mouse trachea into four samples allowed each trachea to yield data at four different timepoints. One trachea sample was immediately imaged as described below (time = 0 min), other samples were subsequently incubated at either 4 °C or room temperature for varying times (30, 60, 120, 360, 720 mins) before being imaged. As each animal provided tissue samples across multiple timepoints no sample randomisation was required. Samples required per group/timepoint (Airway samples from five animals per group) was based off previous studies (*Francis, 2023*; *Scopulovic et al., 2022*). No samples/data were excluded, confounders were not controlled for. Room temperature was maintained by keeping samples in an air-conditioned laboratory at ~24 °C, 4 °C was maintained by placing samples in a 4 °C refrigerator (Pharmaceutical Refrigerator, MPR-721; Panasonic). All samples were imaged at 37 °C following storage. 4 °C or room temperature storage was chosen as they are logically the temperatures most likely encountered by samples during storage/transport to possible imaging locations (37 °C storage/transport not being a common transport option).

**Table 1  Complete list of cell culture media used in study.** Full media formulations are listed in Table S1.

| Cell culture media | Company | Catalogue number |
|---|---|---|
| Saline (0.9% NaCl, no additives) | Aero Healthcare | AW2000LB |
| Dulbecco's Phosphate-Buffered Saline (DPBS) | Sigma-Aldrich | D8662 |
| Hanks' Balanced Salt Solution (HBSS) | Sigma-Aldrich | H1641 |
| Dulbecco's Modified Eagle's Medium (DMEM) | Sigma-Aldrich | D5796 |
| Medium 199 (M199) | Thermo Fisher Scientific | 12350039 |
| Roswell Park Memorial Institute Medium (RPMI) | Thermo Fisher Scientific | 11835030 |
| Minimum Essential Medium (MEM) | Thermo Fisher Scientific | 11575032 |
| Leibovitz's L-15 Medium (L-15) | Thermo Fisher Scientific | 21083027 |

## Cilia imaging

Cilia motility and cilia generated flow was imaged and quantified as previously described (*Scopulovic et al., 2022*). In brief, trachea samples were mounted with ~100 µl of media containing 0.50 µm microspheres (~1 drop per four ml of media; Polysciences Inc; 17152-10) within an imaging chamber constructed using two 24 × 50 mm #1 coverslips (Bio-Strategy; EPBRCS24501GP) sandwiching a ~0.25 mm thick layer of silicone sheet (AAA Acme Rubber Co; CASS-.010X36-64909) from which a central square had been cut to form a shallow walled imaging chamber. Samples were then imaged at 37 °C using Differential Interference Contrast (DIC) microscopy on a Zeiss Axiovert 200 microscope with a 63x/1.4 Oil objective (Zeiss; 420782-9900). The microscope, microscope objective, and objective oil were all kept at 37 °C. Transfer of heat from objective to sample (contained in small drop of media) occurs in seconds due to conduction *via* the immersion oil and small sample volume. ~1 min is taken between samples being placed onto microscope before movies are collected. Previous observations have found this to be sufficient as longer times are not associated with different CBFs, and imaging samples straight from a 37 °C cell culture incubator gives identical CBFs. Recordings of cilia motility were collected using a high speed USB3.0 digital camera with a 1.5MP 1/2.9" Sony Exmore CMOS sensor (EM101500A; ProSciTech), connected to a PC running Windows 10 and ImageView software (ImageView, version x64 4.7). Movie resolution was 0.104 µm/pixel, with overall movie field of view being 1,440 × 550 pixels. One second movies (avi; uncompressed) were collected at ~300 fps for quantification of cilia beat frequency (CBF); 10 s high-efficiency video encoding (HEVC) movies were collected at 30 fps for counting the number of cells with motile cilia per field of view and for quantification of cilia generated flow.

## Quantifying cilia motility and cilia generated flow

The abundance of epithelial cells with motile cilia was first assessed by counting the number of cells with motile cilia visible in each 10 s movie field of view. CBFs were then measured from kymographs generated from cilia motion in the one second movies (~300 fps) using ImageJ (FIJI 2.3.0/1.53q) (*Schindelin et al., 2012*) followed by custom MATLAB script (MathWorks Inc, version 9.9.0, R2020b) as previously described (*Scopulovic et al., 2022*). The average of three cilia kymographs, representing the ciliary motion of three different ciliated cells, were randomly collected per movie. Finally,

Cilia generated flow was quantified by tracking the velocity of microspheres added to the culture media in ten second movies (30 fps) using the Manual Tracking plugin for ImageJ (https://imagej.net/plugins/manual-tracking). Microsphere directionality (ratio representing liner flow) was calculated from the velocity data using Microsoft excel by dividing net microsphere displacement by total distance travelled. Microspheres moving in a straight-line displayed directionality $\approx 1$; while microspheres moving randomly displayed directionality $\approx 0$. Thus, microsphere velocity reflected cilia generated flow speed, while microsphere directionality reflected the linearity of generated flow.

## Data analysis and statistics

Samples were not assessed blindly, but data was quantified automatically *via* MATLAB/ImageJ as described above. Multiple measurements ($\geq 3$) of each parameter (motile cilia abundance, CBF, flow velocity, flow directionality) were made from each movie, with at least two movies collected per tissue sample for each media, temperature, and timepoint, which were then averaged for each movie/animal; resultant group averages (samples from $n = 5$ animals per group) were then compared using two-way ANOVA and *post-hoc* Tukey's multiple comparisons test (Prism 10.3.0; GraphPad Software, San Diego, CA, USA). Data presented as mean $\pm$ SD. $P < 0.05$ was considered significant.

Author declares no potential conflicts of interest. No grant funding was used to conduct this study. No protocol (research question, key design features, or analysis plan) was prepared for this study.

# RESULTS

A total of 225 mice yielding 900 different samples divided into 180 groups were used for this study (see Table S1 for group list; Supplemental File for raw data). In summary, the ability to maintain cilia function over time (0–12 h) was assessed for eight chemical media, following either 4 °C or room temperature storage, in the presence or absence of 10% FBS. Saline was only assessed without FBS (*i.e.,* there were no samples tested in Saline + FBS).

## Direct impact of media on cilia function

The direct impact of each liquid media on respiratory cilia function was assessed by harvesting/dissecting and immediately imaging tracheal tissue within each media (*i.e.,* time = 0; Fig. 1).

Cilia motility was assessed by counting the number of respiratory epithelial cells with motile cilia per microscope field of view (Fig. 1A), and cilia beat frequency (CBF) of those cilia (Fig. 1B). Except for saline, cell counting revealed no significant difference in the number of cells with motile cilia between all media at time 0 mins, with or without FBS. Cell counts ranged between $18.6 \pm 3.8$ cells/field for samples in L15+10% FBS up to $23.8 \pm 2.1$ cells/field for samples in M199 w/o FBS. Saline showed a small but significant reduction in number of observed cells with motile cilia ($12 \pm 4.6$ cells/field) when compared to all other media groups (Fig. 1A). No significant difference was seen in CBF measured between samples in any media group at time = 0 mins, with or without FBS. Observed CBF ranged between $21.7 \pm 3.7$ Hz for cells in Dulbecco's Phosphate-Buffered Saline (DPBS) w/o FBS up to $27.0 \pm 4.1$ Hz for cells in L15+10% FBS (Fig. 1B).

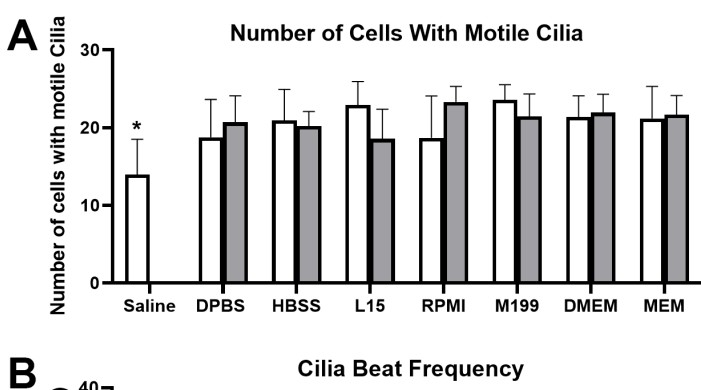

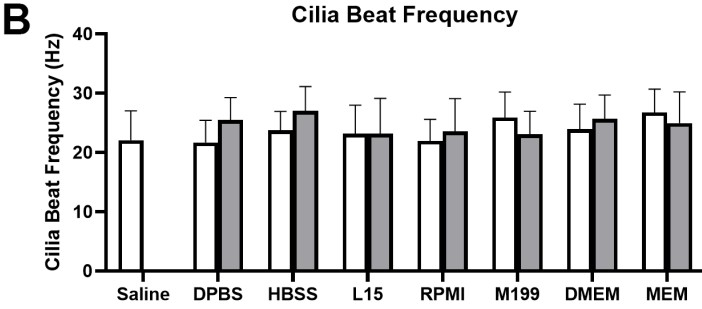

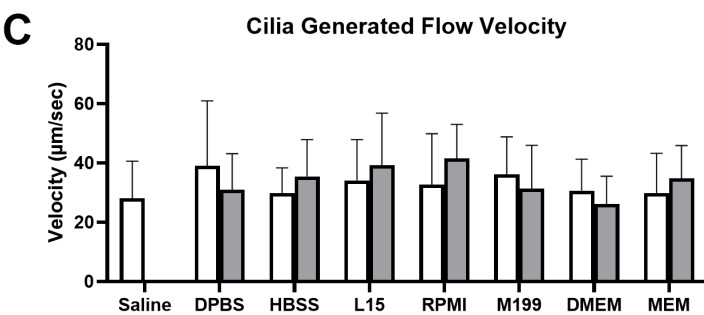

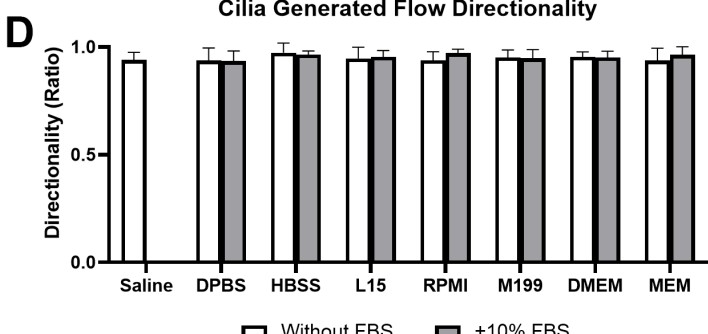

☐ Without FBS    ▧ +10% FBS

**Figure 1   Impact of liquid media on respiratory epithelia cilia motility following tissue harvest and immediate imaging (time = 0 mins).** Cilia motility was assessed by quantifying the number of respiratory epithelial cells with motile cilia per microscope field of view (A), and cilia beat frequency (CBF) of those cilia (B). Cilia generated flow was assessed by tracking microsphere movement within cilia bathing media by quantifying microsphere velocity (C), and microsphere directionality (D). Data is presented as mean ± SD; $n = 10$. * $p < 0.05$ for the saline group *vs* all other groups. NB: Saline was only assessed without FBS.

Cilia generated flow was assessed by tracking microsphere movement within cilia bathing media by quantifying microsphere velocity (Fig. 1C), and microsphere directionality (Fig. 1D). No significant difference was seen in flow velocity between samples in any media group at time = 0 mins, with or without FBS. All samples displayed strong flow across the surface of the ciliated respiratory epithelia (Movie S1), with cilia generated flow velocity ranging between 26.3 ± 9.3 μm/sec for samples in DMEM+10% FBS up to 41.6 ± 11.4 μm/sec for samples in RPMI+10% FBS (Fig. 1C). No significant difference was seen in flow directionality measured between samples in any media group at time = 0 mins, with or without FBS. Directionality is calculated from microsphere movement and indicates the degree of flow linearity, strong cilia generated flow will result in linear flow (microsphere movement) across the surface of the ciliated epithelia giving a directionality ≈ 1; while patchy or random flow (microsphere movement) will result in significant reductions in directionality (<1). All samples in any media showed similar high directionality values (>0.9) corresponding to the generation of strong linear flow across all ciliated epithelia measured, with directionality ranging between 0.93 ± 0.05 for samples in DPBS+10% FBS up to 0.97 ± 0.05 for samples in HBSS w/o FBS (Fig. 1D).

## Impact of sample storage in different media on cilia function

The ability of different media to maintain cilia motility during short term storage (up to 12 h), at either 4 °C or room temperature was subsequently assessed. All data is presented as % change from baseline (time = 0 mins; Fig. 1).

### *Storage media and the general maintenance of cilia motility*

Except for samples stored in saline, cell counting revealed mostly minor differences in the number of cells with motile cilia when stored in all media at all time points, with or without FBS (Fig. 2).

Storage in saline, both at 4 °C or room temperature resulted in a dramatic decrease in the number of cells with motile cilia, which became significantly decreased after only 1 h of storage (−63.7 ± 28% decrease at 4 °C, −42.4 ± 31.4% decrease at room temperature). This decrease continued up to 6 h of storage after which only random isolated cells with motile cilia could be found (if searched for). Storage in 4 °C saline was mildly better than storage at room temperature after 12 h with only a −64.0 ± 24.2% decrease at 4 °C *vs* a −92.1 ± 10.7% decrease at room temperature (Fig. 2) (Movie S2).

Other cell counting results of note include a small but significant elevation from baseline of cell number when stored at room temperate in HBSS + FBS, which became significantly elevated after 2 h storage (+17.3 ± 12%), and remained slightly elevated up to 12 h (+16.3 ± 14.6%). Conversely the 12-hour samples stored in 4 °C HBSS w/o FBS was slightly but significantly reduced (−20.8 ± 17%) (Fig. 2).

The ability of L15 to maintain the number of cells with motile cilia per observed field of view appeared to be significantly impacted by the absence of FBS. Samples stored in L15 w/o FBS for ≥ 2 h displayed significant reductions in cell number following storage at either 4 °C (−16.4 ± 23%) or room temperature (−23.8 ± 31.5). These reductions remained constant up to 12 h storage (−21.9 ± 14.6% at 4 °C; -23.7 ± 13.9% at room temperature) (Fig. 2).

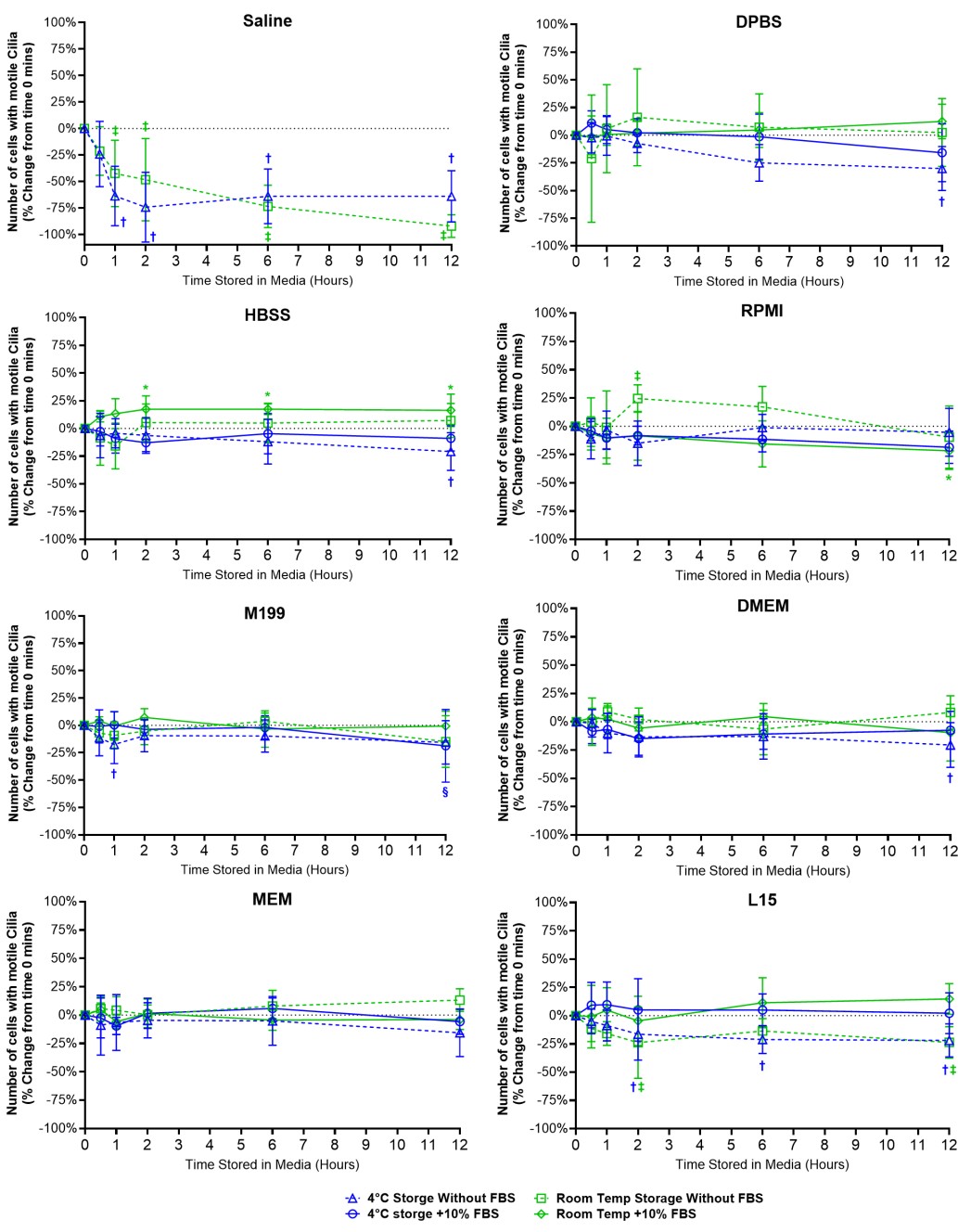

**Figure 2** **Preservation of respiratory cilia function following storage in different liquid media as assessed by quantifying number of respiratory epithelial cells with motile cilia per microscope field of view.** Tracheal samples were harvested and stored in listed media at either 4 °C or room temperature before imaging at either 30, 60, 120, 360, or 720 min. Data is presented as mean ± SD % change from baseline (time = 0 mins; Fig. 1A); $n = 5$ for each timepoint. † storage in 4 °C media without FBS significantly different to baseline ($P < 0.05$), § storage in 4 °C media + 10% FBS significantly different to baseline ($P < 0.05$), ‡ storage in room temperature media without FBS significantly different to baseline ($P < 0.05$), ⋆ storage in room temperature media + 10% FBS significantly different to baseline ($P < 0.05$).

RPMI w/o FBS showed a single significant anomalous elevation following 2 h storage at room temperature (+24.5 ± 12.1%), while 4 °C storage for 12 h caused small significant decreases in cell numbers when stored in M199 + FBS (−18.7 ± 33.2%), and DPBS/DMEM w/o FBS (−30.1 ± 19.9% / −20.5 ± 19.8%) (Fig. 2).

### Storage media and the maintenance of cilia beat frequency (CBF)

A range of different changes in CBF were observed for samples stored in the different media (Fig. 3).

Short term storage in saline (≤ 2 h) resulted in significant reductions in CBF for samples stored at both 4 °C (-43.0 ± 13.5%) and room temperature (38.8 ± 39.1%). This was followed by CBF in 4 °C saline samples returning to baseline levels after 12 h (3.8 ± 35.6%) while CBF in room temperature saline samples dropped by −84.5 ± 18.2% (Fig. 3). It should be highlighted that the CBF results for saline need to be taken in context with the cell count numbers shown in the previous figure (Fig. 2), cell numbers were significantly reduced after short term storage in saline and the CBF results were obtained from the very sparsely remaining ciliated cells that in many cases were hard to find.

One general trend observed for just about all media tested was that room temperature storage of ciliated samples resulted in significant elevations in CBF from baseline values, and this appeared generally more prevalent in media containing FBS (Fig. 3). As seen in DPBS + FBS (+52.5 ± 21.7% after 2 h); HBSS w/o FBS (+63.6 ± 22.0% after 12 h) and HBSS + FBS (+31.4 ± 8.8% after 6 h); RPMI w/o FBS (+45.6 ± 21.3% after 2 h) and RPMI + FBS (+34.7 ± 29.0% after 6 h); M199 + FBS (+28.7 ± 15.3% after 2 h); DMEM + FBS (+26.4 ± 21.1% after 6 h); MEM + FBS (+27.5 ± 15.3% after 6 h); and L15 + FBS (+23.7 ± 11.1% after 2 h).

For just about all media, 4 °C storage was markedly more capable in maintaining tissue CBF at baseline values for all timepoints examined (Fig. 3). One exception being DMEM w/o FBS, where 4 °C storage resulted in a significant elevation in CBF after 6 h storage (+24.7 ± 27.5%). The ability of L15 to maintain CBF in ciliated samples also appeared to rely on the presence of FBS. Samples stored in L15 w/o FBS at both 4 °C and room temperature all showed significant reductions in CBF starting after 1 h storage (−22.6 ± 17.5% at 4 °C; −20.8 ± 20.3% at room temperature) and remaining depressed up to 12 h storage (−41.7 ± 13.3% at 4 °C; −25.7 ± 14.7% at room temperature) (Fig. 3).

### Storage media and the maintenance of cilia generated flow (velocity)

Cilia generated flow velocity was significantly impaired for samples stored in saline (Fig. 4), which was significantly decreased after just 30 mins room temperature storage (−23.9 ± 26.4%) and after 60 min 4 °C storage (−53.1 ± 18.4%). Flow velocity remained significantly decreased at all later timepoints for saline stored samples. As flow velocity was determined by tracking microspheres contained within the media, and not if cilia were generating linear fluid flow across the ciliated epithelial surface, it should be noted that the low velocity values measured from saline samples at later timepoints represented random bead movement by Brownian motion rather than cilia mediated flow (Movie S2) (*Francis, 2023*).

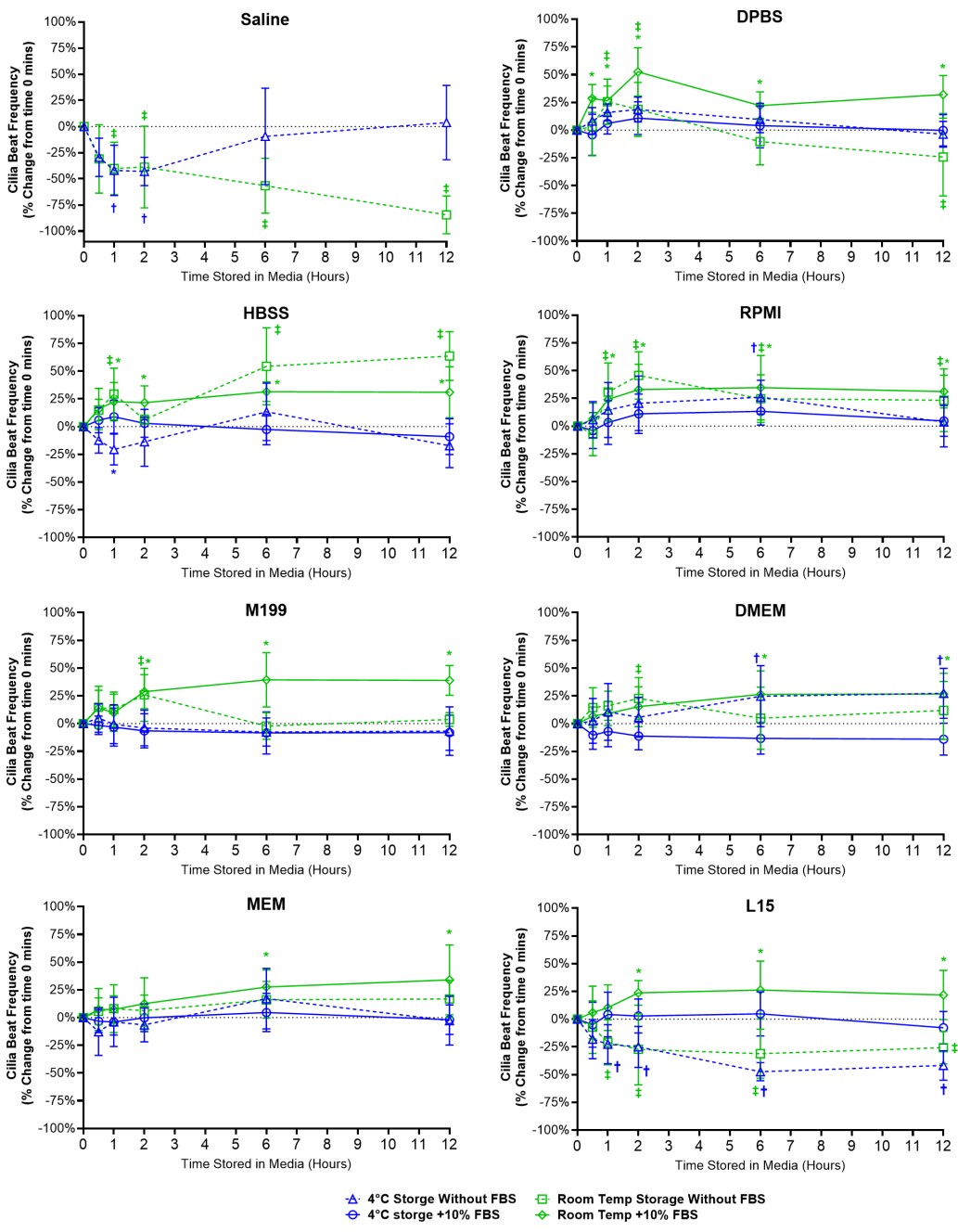

**Figure 3  Preservation of respiratory cilia function following storage in different liquid media as assessed by quantifying cilia beat frequency (CBF).** Tracheal samples were harvested and stored in listed media at either 4 °C or room temperature before imaging at either 30, 60, 120, 360, or 720 min. Data is presented as mean ± SD % change from baseline (time = 0 mins; Fig. 1B); $n = 5$ for each timepoint. † storage in 4 °C media without FBS significantly different to baseline ($P < 0.05$), § storage in 4 °C media + 10% FBS significantly different to baseline ($P < 0.05$), ‡ storage in room temperature media without FBS significantly different to baseline ($P < 0.05$), * storage in room temperature media + 10% FBS significantly different to baseline ($P < 0.05$).

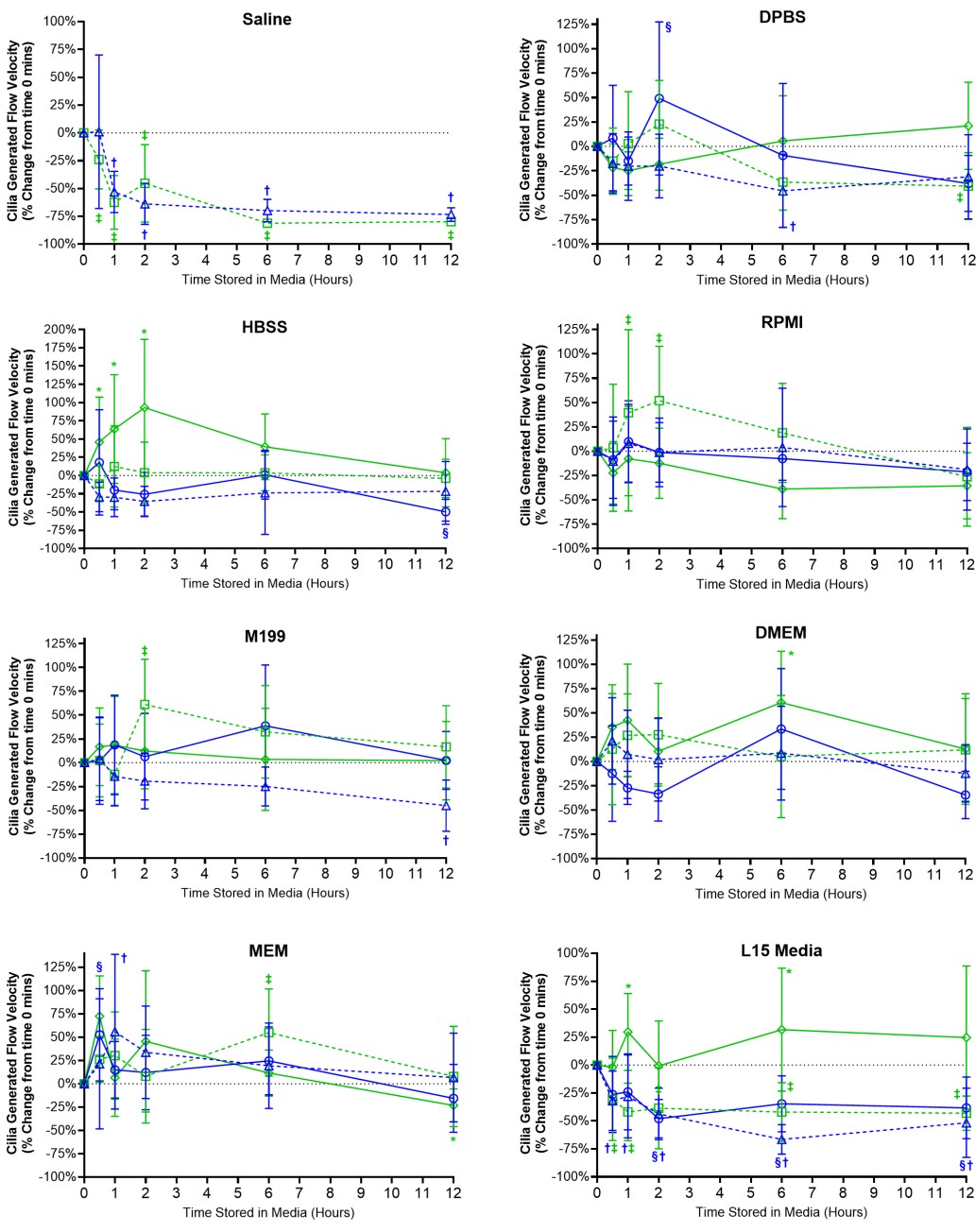

**Figure 4** **Preservation of respiratory cilia generated flow following storage in different liquid media as assessed by quantifying velocity of microspheres within cilia bathing media.** Tracheal samples were harvested and stored in listed media at either 4 °C or room temperature before imaging at either 30, 60, 120, 360, or 720 min. Data is presented as mean ± SD % change from baseline (*i.e.,* time = 0 mins; Fig. 1C); $n = 5$ for each timepoint. † storage in 4 °C media without FBS significantly different to baseline ($P < 0.05$), § storage in 4 °C media + 10% FBS significantly different to baseline ($P < 0.05$), ‡ storage in room temperature media without FBS significantly different to baseline ($P < 0.05$), * storage in room temperature media + 10% FBS significantly different to baseline ($P < 0.05$).

Ciliated samples stored in all other media showed robust maintenance of flow velocity across all time points examined (Fig. 4). However, beyond isolated spikes, some significant trends were observed. Samples stores in room temperature HBSS with FBS displayed significant increases in flow velocity after 30 mins ($+45.8 \pm 61.6\%$) which peaked after 2 h ($93.1 \pm 93.6\%$) before returning towards baseline values after 6–12 h (Fig. 4). A similar significant spike in flow velocity was seen for samples stored in room temperature RPMI w/o FBS at both 1 and 2 h ($+39 \pm 85.3\%$, $+52 \pm 55.9\%$) before returning to baseline values after 6–12 h (Fig. 4). The trend for room temperature storage causing spikes in flow velocity was probably the result of the elevations in CBF also seen in room temperature stored samples (Fig. 3). Significant differences were also seen for room temperature stored samples in M199 w/o FBS after 2 h ($+61 \pm 47.6\%$); DMEM with FBS after 6 h ($+60.7 \pm 52.7\%$); MEM w/o FBS after 6 h ($+55.0 \pm 46.9\%$); L15 media with FBS at the 1 h ($+29.7 \pm 34.4\%$) and 6 h time points ($+31.7 \pm 55.1\%$) (Fig. 4).

The importance of FBS supplementation of L15 was also seen in the velocity data. Samples stored in L15 w/o FBS at both 4 °C and room temperature all showed significant reductions in flow velocities starting after 30 mins storage ($-32 \pm 26.5\%$ at 4 °C; $-30.6 \pm 36.9\%$ at room temperature) and remained depressed up to 12 h storage ($-51.7 \pm 31\%$ at 4 °C; $-43.1 \pm 15.3\%$ at room temperature) (Fig. 4). Of note is that flow velocity was also significantly reduced for samples stored in 4 °C L15 with FBS after 2 h ($-48.0 \pm 17.2\%$), which remained significantly depressed at all subsequent times points up to 12 h ($-38.3 \pm 27.7\%$).

### Storage media and the maintenance of cilia generated flow (directionality)

Directionality is a ratio indicating the degree of flow linearity and provides an easy assessment for the presence or absence of cilia generated flow. Except for samples stored in saline, samples stored in all other media showed little change in directionality indicating the presence of robust linear flow across the surface of all the ciliated epithelia examined (Fig. 5).

Samples in saline showed a significant decrease in directionality after 1 h room temperature storage ($-32.4 \pm 34.3\%$) and after 2 h 4 °C storage ($-57.3 \pm 36.3\%$), and further decreased up to 12 h storage ($-82.9 \pm 13.1\%$ at 4 °C; $-80.5 \pm 10.5\%$ at room temperature) (Fig. 5). The disappearance of linear flow across the surface of saline stored samples would logically arise due to the disappearance of cells with motile cilia in these samples (Fig. 2).

Ciliated samples stored in all other media showed mostly constant directionality. However, beyond isolated small significant outliers, some small but significant trends were observed (Fig. 5). Samples stored in 4 °C HBSS w/o FBS showed small but significant reductions in directionality starting after 30 min storage ($-2.1 \pm 2\%$) which is maintained up to 12 h ($-4.1 \pm 4.1\%$). Samples in room temperature RPMI with FBS showed small but significant reductions in directionality at the 6 and 12 h time points ($-8.1 \pm 5.4\%$, $-4.4 \pm 4.3\%$). Conversely, samples in room temperature DMEM with FBS showed small but significant increases in directionality starting after 30 min storage ($+3.2 \pm 2.0\%$), which was also significant at the 1 and 6 h time points ($+2.9 \pm 2.1\%$, $+3.1 \pm 2.2\%$).

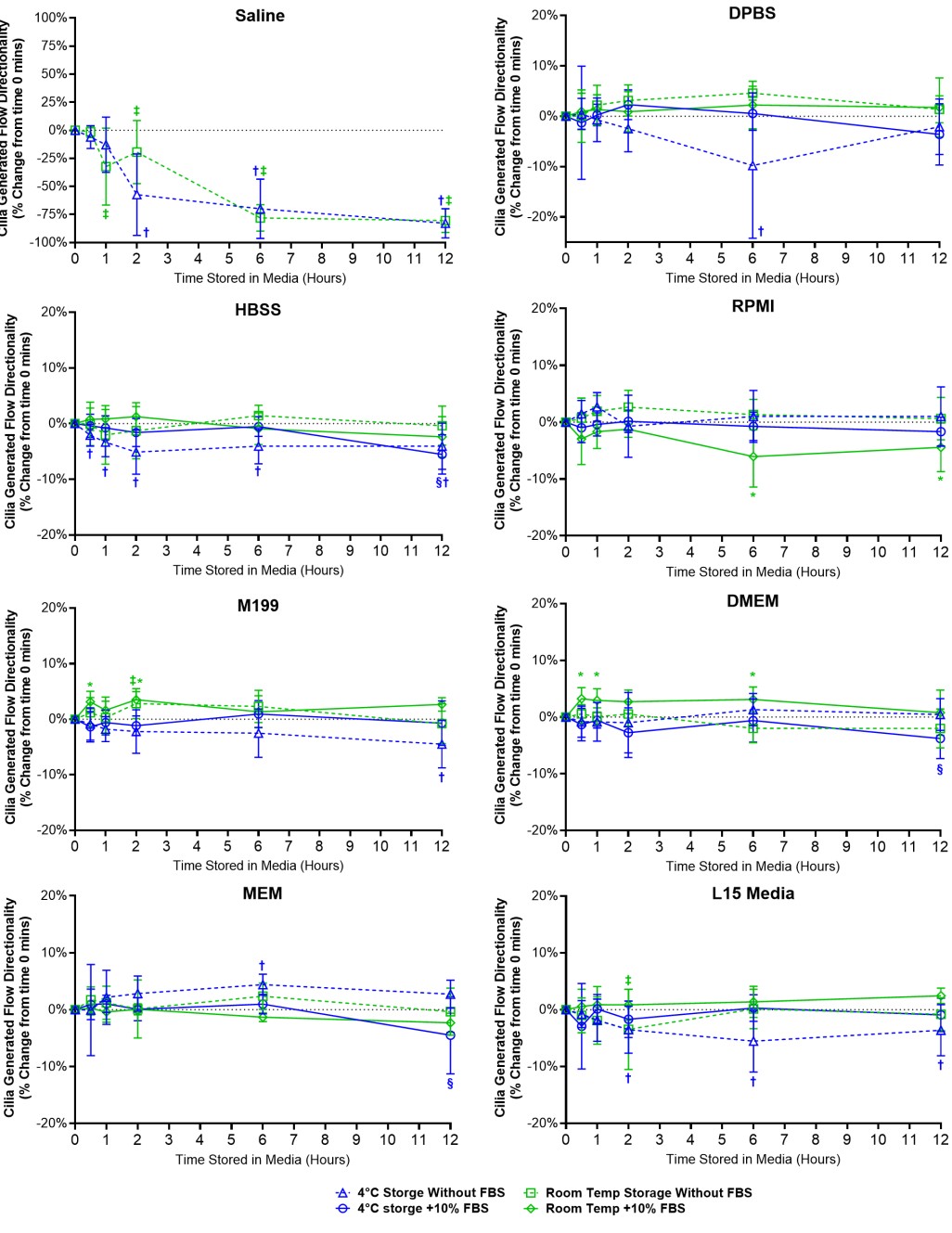

**Figure 5  Preservation of respiratory cilia generated flow following storage in different liquid media as assessed by quantifying directionality of microspheres within cilia bathing media.** Tracheal samples were harvested and stored in listed media at either 4 °C or room temperature before imaging at either 30, 60, 120, 360, or 720 min. Data is presented as mean ± SD % change from baseline (*i.e.,* time = 0 mins; Fig. 1D); $n = 5$ for each timepoint. † storage in 4 °C media without FBS significantly different to baseline ($P < 0.05$), § storage in 4 °C media + 10% FBS significantly different to baseline ($P < 0.05$), ‡ storage in room temperature media without FBS significantly different to baseline ($P < 0.05$), ⋆ storage in room temperature media + 10% FBS significantly different to baseline ($P < 0.05$).

Samples in room temperature M199 with FBS also showed small but significant increases in directionality at the 30 min and 2 h time points (+3.1 ± 1.9%, +3.5 ± 1.5%). Finally, Samples in 4 °C L15 w/o FBS showed small but significant reductions in directionality starting after 2 h storage (−3.5 ± 4.1%) which is maintained up to 12 h (−3.6 ± 4.5%).

## DISCUSSION

The aim of this study was to assess the role of liquid media, FBS supplementation, and storage time on the short-term maintenance of motile cilia function in isolated respiratory epithelia samples. Motivation for this study is due to the large variability in reported control cilia motility (*e.g.*, CBF) present in the literature (*Fawcett et al., 2023*; *Price et al., 2015*; *Scopulovic et al., 2022*; *Yasuda et al., 2020*; *Zahid et al., 2020*). While past cilia variability may be related to experimental model used (animal *vs* human; fresh *vs* cultured tissue), the influence played by the liquid media used to harvest, suspend, and image tissue should not be overlooked. While this study was designed to mainly assess the short-term storage of tissue one expects to find within a typical laboratory, 4 °C and room temperature storage was investigated in preference of 37 °C storage to better mimic possible conditions faced by tissue during transport outside the laboratory. The goal being to also provide data for standardization of ciliated tissue handling protocols needed for the collection and transport of ciliated samples to specialized laboratories for possible diagnostic testing (*e.g.*, PCD) (*Despotes et al., 2024*; *Shapiro et al., 2018*).

### Cilia motility in freshly harvest tissue

The initial results obtained using samples immediately imaged following harvest ($t = 0$ mins) show that cilia in all media displayed regular, fast beating, and were able to generate fast linear flow across the surface of the respiratory epithelia in line with previous studies (Movie S1) (*Francis, 2023*; *Scopulovic et al., 2022*). There were no notable or significant differences in cilia motility parameters such as CBF and cilia generated flow for samples in any cell culture media (Fig. 1). Indeed, even plain unsupplemented 0.9% saline appeared capable of maintaining most cilia function in the very short term, with only a small but significant decrease in ciliated cell numbers. Similarly, supplementation of cell culture media with FBS also appeared to have no significant effect (negative or positive) on any cilia motile parameter measured.

### Cilia motility following storage

Cilia motility was also well maintained in all liquid media (except saline) following both room temperature and 4 °C short-term storage (0–12 h), with or without FBS supplementation (Movie S2).

#### *Saline vs DPBS*

Unsurprisingly, saline was not able to maintain cilia function, with large and significant decreases in all cilia functional parameters seen following storage of samples in saline (Figs. 2–5). As saline contains no nutrients/growth factors, saline storage was expected to negatively impact cell viability and thus cilia function (*Chen et al., 2017*). However, the impact of this effect was fast and dramatic with decreases in all motile parameters

after just 30 min in saline (becoming significant after just 1 h). It was also interesting to note that 4 °C storage did nothing to help prevent this degradation of cilia function in saline, suggesting saline should not be used for assessment of cilia function except when immediately imaging fresh tissue. Conversely DPBS, while only slightly more complex in composition than saline (but also lacking nutrients/growth factors), appeared very capable in maintaining cilia motility during storage. As highlighted by ciliated samples, even when stored for up to 12 h in room temperature DPBS, still displaying cilia motility on par with that found in samples stored in more complex media. The difference between saline and DPBS for maintaining cilia motility indicates that a relatively simple factor is essential for maintaining cell viability/cilia function in these samples, such as the presence of calcium, magnesium, or phosphate buffering of pH changes, future studies are needed to pinpoint what mechanism is critical. The finding that DPBS can maintain ciliated tissues and cilia function for up to 12 h will however prove very useful as it provides a relatively 'blank canvas' for future studies exploring factors regulating cilia function by removing a lot of extraneous variables from the experimental equation (*i.e.,* the large number of different factors present in more complex cell culture media formulations not to mention those found in FBS).

### Room temperature storage causes increased cilia motility

While robust cilia motility and cilia generated flow was maintained for up to 12 h in all media more complex than saline, some variations in cilia function were observed between sample groups. Most notable was the observation that for many samples room temperature storage results in significant elevations of cilia motility (CBF and cilia generated flow) compared to samples stored at 4 °C. These findings match previous observations from freshly harvested human respiratory epithelia, where increased CBF was seen after: storage in 22 °C RPMI media (*Sommer et al., 2010*), room temperature storage in M199 media (*Bricmont et al., 2023*) and room temperature storage in DMEM (when compared to 4 °C storage) (*Reula et al., 2021*).

Temperature is well known to influence CBF in a range of cilia models, with increased temperature causing a corresponding linear increase in CBF (*Chen, Lemieux & Wong, 2016*; *Christopher et al., 2014*; *Schipor et al., 2006*). However, a role for storage temperature later impacting cilia motility when imaged at 37 °C has not been previously acknowledged. It may be hypothesized that whereas cells stored at 4 °C remains relatively static, room temperature storage causes cells to develop abnormal cellular physiology, which may be related to abnormal enzymatic function(s) inherent with suboptimal temperatures (*Knapp & Huang, 2022*). More work is needed to determine if this elevation in CBF is permanent or may disappear if cells are given more time at 37 °C to recover from storage. However, even with this elevation in CBF following room temperature storage, ciliated samples in these groups still displayed robust cilia motility and cilia generated flow. Thus, room temperature storage may still be a viable transport option when looking to standardize handling protocols for ciliated sample shipping to facilities for functional diagnosis of cilia function. Such elevations in CBF may not be expected to influence diagnostic testing of

diseases characterized by dramatic defects in cilia motility, such as PCD where just about all cilia motility is lost (*Despotes et al., 2024*; *Shapiro et al., 2018*).

### FBS supplementation not essential except when using L15

Another important observation was that except for L15, FBS supplementation appears relatively unimportant for the maintenance of cilia motility when stored in the different media. The finding that FBS supplementation was essential for maintaining cilia function in L15 was unexpected, was found for samples stored at both 4 °C or room temperature, and was characterized by significant reductions in ciliated cell abundance, CBF, and cilia flow velocity.

L15 is a commonly used media for mammalian cell experiments as it doesn't require $CO_2$ for pH control, so the finding that L15 without FBS was significantly worse than other media in maintaining cilia function was surprising and has not been previously reported. One noticeable difference between L15 and other media is its energy source (carbohydrate), L15 is the only media lacking glucose and instead contains galactose (Table S1). Glucose and galactose are epimers which differ in the position of one hydroxyl group, and enter glycolysis through different routes (*Chandel, 2021*). L15 contains galactose because it's metabolized more slowly and doesn't produce as many acidic by-products than glucose (*Conte et al., 2021*), which helps L15 maintain a stable pH in the absence of $CO_2$ buffering. No significant adverse effects of L15 using galactose instead of glucose as an energy source has been previously reported for mammalian cell studies. However, it has been shown that yeast cells display altered metabolism and metabolic gene expression if grown in galactose rich environments (*Escalante-Chong et al., 2015*). Thus, it's possible that similar changes in metabolism and gene expression could also occurs in mammalian cells when the dominant energy source is switched from glucose to galactose. This could severely impact ciliated cells which have high energetic requirements (*Villar, Vergara & Bacigalupo, 2021*) resulting in the observed reduction in cilia motility when the tissue was stored in L15 alone. Supplementation with FBS, which contains glucose (108–166 mg/dL) (*Lee et al., 2023*) could prevent this altered metabolism and thus maintains cilia motility. Such a hypothesis requires more work but may indicate a major issue with L15, especially if used without FBS with cell cultures that have high energetic requirements, and this may impact other cellular processes beyond simple cilia motility.

### Other possible effects impacting cilia motility

Other factors that differ for the media studied which are known to influence cilia motility include osmotic stresses, pH changes, and possible chemosensory stimulation. While the osmolarity of the different media studied are necessarily similar by design, due to differences in formulations they may not be identical, and evidence exists for osmolarity influencing CBF. Elevated osmolarity reduced CBF in freshly isolated human and murine samples (*Horvath & Sorscher, 2008*) while hypertonic nebulized saline reduced CBF in isolated sheep trachea (*Kelly et al., 2023*). However, these osmolarity studies utilized much greater changes in osmolarity than one can expect to find between the different media used in this study, so more work would be required to determine if it plays a role here. pH is also well known to impact CBF with elevations in pH resulting in significant elevations

in CBF (*Sutto, Conner & Salathe, 2004*), the different media in this study utilize a range of methods to buffer pH changes, meaning it's possible that pH changes are also responsible for some of the changes observed. Finally, cilia are recognized to be chemosensory (*Shah et al., 2009*), thus it cannot be ruled out that the different media formulations may result in different cilia chemosensory effects causing subsequent changes in cilia motility.

## STUDY LIMITATIONS

One limitation of this study is the varying amounts of energy (glucose or galactose) contained within the different media examined, making direct comparison of media effects on cilia motility more difficult. 4 °C storage is expected to minimize this issue by significantly retarding cell metabolism and resultant energy requirements, while room temperature storage is expected to still allow cell metabolism, albeit reduced, to proceed. These suppositions could explain why CBF is maintained following 4 °C storage in DPBS which contains no energy source, while CBF shows a small slow decline in room temperature DPBS after 6–12 h as energy stores are slowly expended. Conversely, the idea that increased concentrations of energy (glucose) may be associated with increased cilia motility cannot be ruled out and needs to be assessed in future studies directly targeted at understanding the role of glucose in modulating airway cilia motility.

Another possible limitation of this study is the possibility that media and/or storage may cause sample degradation. I attempted to control for this by manual counting of ciliated cell number in each sample, but this may not reflect more subtle cell/tissue injury. To properly test for this, live/dead staining and or apoptosis staining will be needed.

As with any data collected using animal models, further work will be necessary to confirm the findings of this study with human samples, especially for the standardization of handling protocols for diagnostic purposes. However, as mouse models are extensively used to study cilia motility and mucociliary clearance within the airways (*Bustamante-Marin & Ostrowski, 2017*), this data should prove invaluable for future studies.

## CONCLUSIONS

In conclusion, the main findings of this study include firstly, that most common liquid media including DPBS display a similar ability to maintain cilia motility in respiratory tissue samples when stored at 4 °C (for up to 12 h). Secondly, storage of respiratory tissue at room temperature may cause increased CBF, especially if the storage media contains FBS. Finally, only L15 media appears to require FBS supplementation to maintain cilia motility, with cells in L15 lacking FBS displaying reduced CBF following both 4 °C and room temperature storage. The results of this study will be useful for helping standardize handling protocols for the collection/storage/transport of ciliated tissue samples for study and diagnostic purposes.

## ACKNOWLEDGEMENTS

The author wishes to thank Christine Hall for her technical support, and Serrin Rowarth and her team at the Australian Institute of Tropical Health & Medicine Small Animal Colony for providing the animals used in this study.

### Funding

The author received no funding for this work.

### Competing Interests

The author declares that they have no competing interests.

### Author Contributions

- Richard Francis conceived and designed the experiments, performed the experiments, analyzed the data, prepared figures and/or tables, authored or reviewed drafts of the article, and approved the final draft.

### Animal Ethics

The following information was supplied relating to ethical approvals (i.e., approving body and any reference numbers):

James Cook University Animal Ethics Committee (Ethics# A2783).

### Data Availability

The raw measurements are available in the Supplemental File.

The movies are available in the Supplemental Files and figshare:

- Francis, Richard (2024). Supplementary Movie 1. Representative movies of ciliated mouse tracheal epithelia harvested and dissected/isolated in different cell culture media (without FBS) then immediately mounted and imaged using each media. Particles observed in media are 0.50 $\mu$m microspheres added for cilia generated flow quantification. Scale Bar = 10 $\mu$m. figshare. Media. https://doi.org/10.6084/m9.figshare.26878474.v1.

- Francis, Richard (2024). Supplementary Movie 2. Representative movies of ciliated mouse tracheal epithelia harvested, dissected/isolated, and stored at 4 °C for 12 hours in different cell culture media (without FBS) before mounting and imaging (at 37 °C) in each media. Particles observed in media are 0.50 $\mu$m microspheres added for cilia generated flow quantification. Scale Bar = 10 $\mu$m. figshare. Media. https://doi.org/10.6084/m9.figshare.26878555.v1.

### Supplemental Information

Supplemental information for this article can be found online at http://dx.doi.org/10.7717/peerj.19191#supplemental-information.

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
