# Peer review of "Assessment of liquid media requirements for storing and evaluating respiratory cilia motility"

_PeerJ, doi:10.7717/peerj.19191_

## Round 0.1 · original submission · Major Revisions

As you will see, both reviewers recognised the validity of your study but have raised a number of points, each of which should be carefully and fully addressed in a revised paper.

Please address all issues raised, and ensure that ALL data from ALL technical and biological replicates is uploaded to the journal web site; this is essential.

Reviewer 1 ·

Basic reporting

The manuscript is clearly written, but contains several spelling mistakes and incorrect punctuations that should be corrected. Further, the author classifies saline, Dulbecco’s phosphate-buffered saline, and Hanks’ balanced salt solution as cell culture media, but these solutions lack most of the components required for cell cultivation. The purposes of these solutions are mainly to transport, to store or to wash cells but not to culture them.

Some references used in the introduction should be revised in the following three cases.
The author describes in the introduction the problem that “cilia studies” reported large variation in control CBF ranging between 7 and 25 Hz. The cited literature, however, reported differences between 7 and 15 Hz in control CBF. This makes the problem described hardly comprehensible. The author should cite literature demonstrating the mentioned variation.

Further, the citation of Awatade et al. 2023 is inappropriate in the specific context. The author states that this study showed a 50% change in CBF depending on the medium used. However, in fact two different media were used to differentiate the cells into ciliated epithelium. The CBF was determined from the resulting differentiated cells, which obviously had different properties due to differences in the differentiation. One of the used medium is known to drive less efficient differentiation into ciliated epithelial cells.

And Kelly et al. 2023, did not use saline per se but nebulized saline to study the hydration effect on mucociliary transport when saline is used as carrier for nebulized drugs.

The structure of the article conforms to an acceptable format. The figures are relevant to the content of the article. In figure 2 – 5, the mean and SEM are shown, whereas it seems that in the manuscript text, the standard deviation is mentioned instead. This is confusing.

A table with raw data has been made available. However, this table raises some questions.
The author used DPBS without Ca²+/Mg²+ additionally, but did not report on these data in the manuscript. What about these data? In the table is written that the author used M199 medium with HBSS, but this is not mentioned in the manuscript. This should be clarified in the methods part at least. Further, the author states that 5 animals were used for each time point and condition and that multiple measurements were made. Looking at the raw data table, this is not comprehensible. How are the individual values related?

Experimental design

The manuscript clearly defines the research question (how media, storage time and temperature influence cilia function). Cilia function is also known to dependent on different other parameters e.g. temperature, age, osmotic and mechanical stress, and anatomical location of the sample. Especially the factors temperature, age, and osmotic stimulation seem to be relevant in this study and are not addressed sufficiently. The author should give more information about the age of the mice used. Further, for how long were the samples adjusted to 37°C before the measurements?
Sommer et al. 2010 discussed that the RPMI medium composition may cause osmotic and chemosensory stimulation causing increased CBF. This point could be considered in the discussion rather than the impact of the storage temperature.

Validity of the findings

One limitation of this study is that the media, which were used, contain different concentrations of the energy source glucose or galactose ranging from 0 to 4500 g/L. Considering that the beating of the cilia is an energy-intensive process, a direct comparison of the media is difficult, especially after several hours of storage. This point needs further discussion.
Furthermore, the comparison of the CBF measurements with other studies that used controlled or even uncontrolled temperatures like room temperature, 22°C or 37°C for the measurements is very difficult considering the temperature dependence of the CBF. In addition, the author must further consider the degradation of the samples during storage depending on the storage solution used.
Unfortunately, the question raised in the abstract “how cell culture media influence cilia functions” stays unanswered. The specific impact of the solution compositions is hardly discussed.

Reviewer 2 ·

Basic reporting

It would be helpful if the author could add a break in the Y axis in Figure 5 to expand the data between 0 and +/-10% for all the media except saline. This would allow better visualization of the data in that range.

Experimental design

It is indicated that all samples were imaged at 37°C following storage at 4°C or room temperature. But were the samples transferred to pre-warmed media/buffers or were they directly placed in a 37°C chamber? This detail should be included in the Methods.

Validity of the findings

The data demonstrate that saline significantly affects cilia motility, whereas DPBS does not. In the discussion, the author concludes that this finding "indicates that a relatively simple factor is essential for maintaining cell viability/cilia function in these samples, future studies will be needed to pinpoint what this factor is". A plausible explanation is the presence of calcium and magnesium in DPBS, as these essential ions not only help maintain tight junction integrity and epithelial structure but also play a critical role in regulating cilia beating. This should be included in the discussion.

Additional comments

Results line 242: …impacted by “absence” of FBS. Instead of “presence” might be more appropriate.
Discussion line 338: it looks like a word is missing. …which is highlighted in a recent “paper” which showed…

---

## Round 0.2 · accepted · Accept

I am of the opinion that all issues highlighted in review have been addressed and thus I am happy to recommend acceptance.